# Epidermal barrier defects link atopic dermatitis with altered skin cancer susceptibility

Sara Cipolat[1,2†], Esther Hoste[1,2†], Ken Natsuga[2,3], Sven R Quist[2,4], Fiona M Watt[1*]

[1]Centre for Stem Cells and Regenerative Medicine, King's College London, London, United Kingdom; [2]Cancer Research UK Cambridge Research Institute, Cambridge, United Kingdom; [3]Department of Dermatology, Hokkaido University, Sapporo, Japan; [4]Department of Dermatology and Venereology, Otto von Guericke University Magdeburg, Magdeburg, Germany

**Abstract** Atopic dermatitis can result from loss of structural proteins in the outermost epidermal layers, leading to a defective epidermal barrier. To test whether this influences tumour formation, we chemically induced tumours in EPI−/− mice, which lack three barrier proteins—Envoplakin, Periplakin, and Involucrin. EPI−/− mice were highly resistant to developing benign tumours when treated with 7,12-dimethylbenz(a)anthracene (DMBA) and 12-O-tetradecanoylphorbol-13-acetate (TPA). The DMBA response was normal, but EPI−/− skin exhibited an exaggerated atopic response to TPA, characterised by abnormal epidermal differentiation, a complex immune infiltrate and elevated serum thymic stromal lymphopoietin (TSLP). The exacerbated TPA response could be normalised by blocking TSLP or the immunoreceptor NKG2D but not CD4+ T cells. We conclude that atopy is protective against skin cancer in our experimental model and that the mechanism involves keratinocytes communicating with cells of the immune system via signalling elements that normally protect against environmental assaults.

**\*For correspondence:** fiona. watt@kcl.ac.uk

†These authors contributed equally to this work

**Reviewing editor**: Elaine Fuchs, Rockefeller University, United States

## Introduction

There is an ongoing debate as to whether allergic disease is a risk factor for cancer or whether it is protective, with recent studies indicating that the relationship is complex and site specific (*Arana et al., 2010*; *Wedgeworth et al., 2011*; *Hwang et al., 2012*). Several epidemiological studies have suggested that atopic dermatitis (AD; eczema), hives, allergies to animal fur, and certain food ingredients are inversely associated with cancers of tissues that provide an interface with the external environment, such as the skin (*Jensen-Jarolim et al., 2008*; *Sherman et al., 2008*). In a recent large-scale analysis, the risk of nonmelanoma skin cancer was reduced in patients who had both allergic rhinitis and asthma (*Hwang et al., 2012*).

It is challenging to draw firm conclusions from the epidemiological data, in part because of the relapsing and remitting nature of AD and the potential cancer modulatory effects of treatments that are used to manage AD. Therefore, we sought to examine the link between AD and cancer susceptibility using a mouse model of AD in which the primary defect is in the epidermal barrier that is normally protective against pathogens. The skin barrier is formed by terminally differentiated keratinocytes in the outermost layers of the epidermis, known as the cornified layers or stratum corneum. In cornified keratinocytes, the plasma membrane is replaced with a layer of highly insoluble, transglutaminase cross-linked proteins with covalently attached lipids, known as the cornified envelope (*Candi et al., 2005*). Involucrin, envoplakin, and periplakin are the first proteins to be cross-linked by transglutaminase-1 and create the protein scaffold for the attachment of lipids on

**eLife digest** Skin cancer is a common and growing problem—according to the World Health Organization, skin cancers account for one in every three cancers diagnosed world wide. There is some evidence from epidemiological studies that patients with certain allergies might be protected against cancer and, in particular, that the allergic skin condition atopic dermatitis is associated with reduced levels of various skin cancers. However, it is difficult to know if this reduction is due to the atopic dermatitis itself or to the drugs used to treat this allergy.

Genetically engineered mice that are lacking three proteins that are involved in the formation of the cornified envelope—the protective layer that replaces the normal plasma membrane in the cells of the outermost skin layers—can be used to study atopic dermatitis. These 'triple knockout mice' have a defective epidermal barrier and altered levels of immune T-cells in the skin.

Now Cipolat et al. have investigated whether defects in the epidermal barrier protect against skin cancer. Knockout mice and wild-type mice were treated with two chemicals: DMBA, which causes mutations in a gene called *HRas*, and TPA, which promotes the formation of tumours from cells that contain *HRas* mutations. After about 16 weeks almost all of the wild-type mice had at least one benign tumour, whereas half of the knockout mice had no tumours. Overall, the average number of benign tumours per mouse was six times higher in the wild-type mice. This shows that the mutations that cause the epidermal barrier defects in knockout mice also protect them against the tumours caused by the combined effects of DMBA and TPA.

Cipolat et al. then compared how the mice responded to DMBA or TPA alone. The knockout mice and the wild-type mice responded to DMBA in the same way; however, the knockout mice showed an exaggerated response to TPA, including a strong inflammatory reaction. This response comprised the production of higher levels of various proteins that are involved in communications between skin cells and the immune system. Cipolat et al. propose that the immune reaction caused by this exaggerated response could help to prevent tumour formation by eliminating tumour-forming cells in the skin.

which the cornified envelope assembles (*Rice and Green, 1977*; *Simon and Green, 1984*; *Ruhrberg et al., 1996*, *1997*).

Mice triply deficient in *envoplakin*, *periplakin,* and *involucrin* (EPI−/− mice) have a defective epidermal barrier and exhibit a reduction in epidermal γδTCR+ CD3+ cells (dendritic epidermal T cells; DETCs) and infiltration of CD4+ T cells into the dermis (*Sevilla et al., 2007*). In contrast, the individual and double knockouts for *envoplakin*, *periplakin,* and *involucrin* do not have observable barrier defects or an altered immune infiltrate (*Sevilla et al., 2007*). The skin barrier phenotype of EPI−/− mice shares similarity with the *flaky tail* mouse, which has nonsense mutations in the keratin filament associated protein filaggrin and the transmembrane protein Tmem79, which is a component of lamellar granules (*Fallon et al., 2009*; *Sasaki et al., 2013*; *Saunders et al., 2013*), the latter resulting in defective stratum corneum formation. Filaggrin mutations in humans are linked to ichthyosis vulgaris (dry, flaky skin) and increased risk of AD (*Palmer et al., 2006*, *2007*; *Brown and McLean, 2012*), while Tmem79 mutations are found in some patients with AD (*Saunders et al., 2013*).

To test the potential association between a defective skin barrier, AD and cancer, we have subjected EPI−/− mice to the classic two-stage chemical carcinogenesis protocol (*Perez-Losada and Balmain, 2003*). A single exposure to the polycyclic aromatic hydrocarbon 7,12-dimethylbenz(a) anthracene (DMBA) induces oncogenic mutations in HRas (initiation). Repeated applications of the promoting agent 12-O-tetradecanoylphorbol-13-acetate (TPA) (promotion) allow initiated, mutagenized cells to clonally expand and form benign tumours called papillomas, some of which progress to malignant squamous cell carcinomas (SCCs) (*Abel et al., 2009*). While most human skin SCCs are associated with UV, rather than chemical, carcinogenesis, 10% do have Ras gene mutations (*Lopez-Pajares et al., 2013*), and DMBA/TPA carcinogenesis has contributed substantially to our understanding of the steps of tumour development in human skin (*Hirst and Balmain, 2004*).

# Results

## EPI −/− mice are resistant to chemical carcinogenesis

To determine whether lack of *Evpl*, *Ppl* and *Ivl* influences sensitivity to tumour formation, we compared DMBA/TPA skin carcinogenesis (*Abel et al., 2009*) in mice lacking all three genes (EPI−/−) and wild-type (WT) mice on the same genetic background. Since EPI−/− mice were on a mixed Sv129 and C57Bl/6 background, F2 crosses between WT Sv129 and C57Bl/6J mice were used to generate the WT controls for all experiments. The data from one experiment are presented in *Figure 1*. A second, independent, carcinogenesis experiment gave similar results (data not shown). Mice that received DMBA or TPA alone did not develop tumours.

The first papillomas emerged in both groups 5–6 weeks after the start of TPA promotion, and the maximum number of papillomas was reached by 15–16 weeks (*Figure 1A*). The average number of papillomas per mouse was approximately sixfold lower in EPI−/− than WT mice (*Figure 1A*; week 15, p=0.0007; week 57, p=0.0001). The incidence of papilloma formation was also lower in EPI−/− mice (*Figure 1B*). Over 95% of WT mice had at least one papilloma 14 weeks after the start of promotion, whereas 50% of EPI−/− mice were tumour-free at the end of the experiment (p=0.0063). Thus combined deletion of *envoplakin*, *periplakin*, and *involucrin* decreased epidermal susceptibility to chemically induced benign tumours.

As expected from the genetic background of the mice (*Woodworth et al., 2004*), only a small subset of papillomas progressed to invasive squamous cell carcinomas (SCCs) (*Figure 1C*). There was no difference in SCC burden or incidence between EPI−/− and WT mice (*Figure 1C,D*). However, since EPI−/− mice had fewer papillomas, the SCC conversion frequency was higher than in WT mice. We observed a downregulation of envoplakin, periplakin, and involucrin in WT papillomas and SCCs (*Figure 1E*), leading us to speculate that the increased risk of SCC conversion in EPI−/− papillomas may be linked to the lack of two desmosomal proteins (*Thiery and Chopin, 1999*). No metastases were observed in any mice (data not shown).

## DMBA responsiveness is unaffected in EPI−/− mice

To determine whether the resistance of EPI−/− mice to papilloma formation was due to a reduced DMBA response, we examined responses to a single topical application of carcinogen. Skin was analysed 24 hr after one application of DMBA in 4 mice per genotype.

DMBA binds to the aryl hydrocarbon receptor (AhR), leading to upregulation of the cytochrome P-450 enzymes Cyp1a1 and Cyp1b1, which convert DMBA into the active metabolite that causes *HRas* mutations (*Gao et al., 2005*). There were no differences in expression of AhR, the AhR repressor (AhRR), Cyp1a1 and Cyp1b1 between WT and EPI−/− epidermis, either in the steady state or after application of DMBA (*Figure 2A*). Langerhans cells resident in the epidermis metabolise DMBA to the active form and as a result DMBA/TPA treated Langerhans cell-deficient mice do not develop papillomas (*Modi et al., 2012*). However, EPI−/− and WT epidermis contained similar numbers of CD207[+] Langerhans cells (*Figure 2B*). We conclude that DMBA was activated to a similar extent whether or not the epidermal cornified envelope was defective.

We next analysed epidermal protective responses to carcinogen-induced DNA damage (DNA repair and induction of apoptosis). Phosphorylated Histone H2A variant H2AX (γH2AX) is recruited to sites of dsDNA damage and subsequent breaks, and is upregulated on DMBA treatment. The proportion of epidermal cells that expressed γH2AX was similar in EPI−/− and WT epidermis, whether or not the skin had been treated with DMBA (*Figure 2C*). The proliferative response of keratinocytes to DMBA was comparable in both genotypes, as assessed by the proportion of phospho-Histone 3 (PH3) positive cells (*Figure 2D*). Furthermore, topical application of DMBA resulted in a similar apoptotic response, as detected by cleaved caspase-3-positive cells and quantitation of mRNA levels for a panel of apoptotic and anti-apoptotic genes (*Figure 2E,F*).

Levels of the DMBA 'signature' *Hras* (codon 61) mutation (*Modi et al., 2012*) were the same in mRNA extracted from SCCs of WT or EPI−/− mice (*Figure 2G*). We did not detect mutations in exons 4, 5, 8 and 9 of *p53* in WT or EPI−/− tumours (data not shown).

We conclude that differences in DMBA uptake and activation or activity did not account for the resistance of EPI−/− mice to chemical carcinogenesis.

## EPI−/− mice exhibit an exacerbated keratinocyte response to TPA

Since WT and EPI−/− mice did not differ in DMBA responsiveness, we next investigated whether their response to the promotion phase of the two-stage carcinogenesis protocol was aberrant. Mice

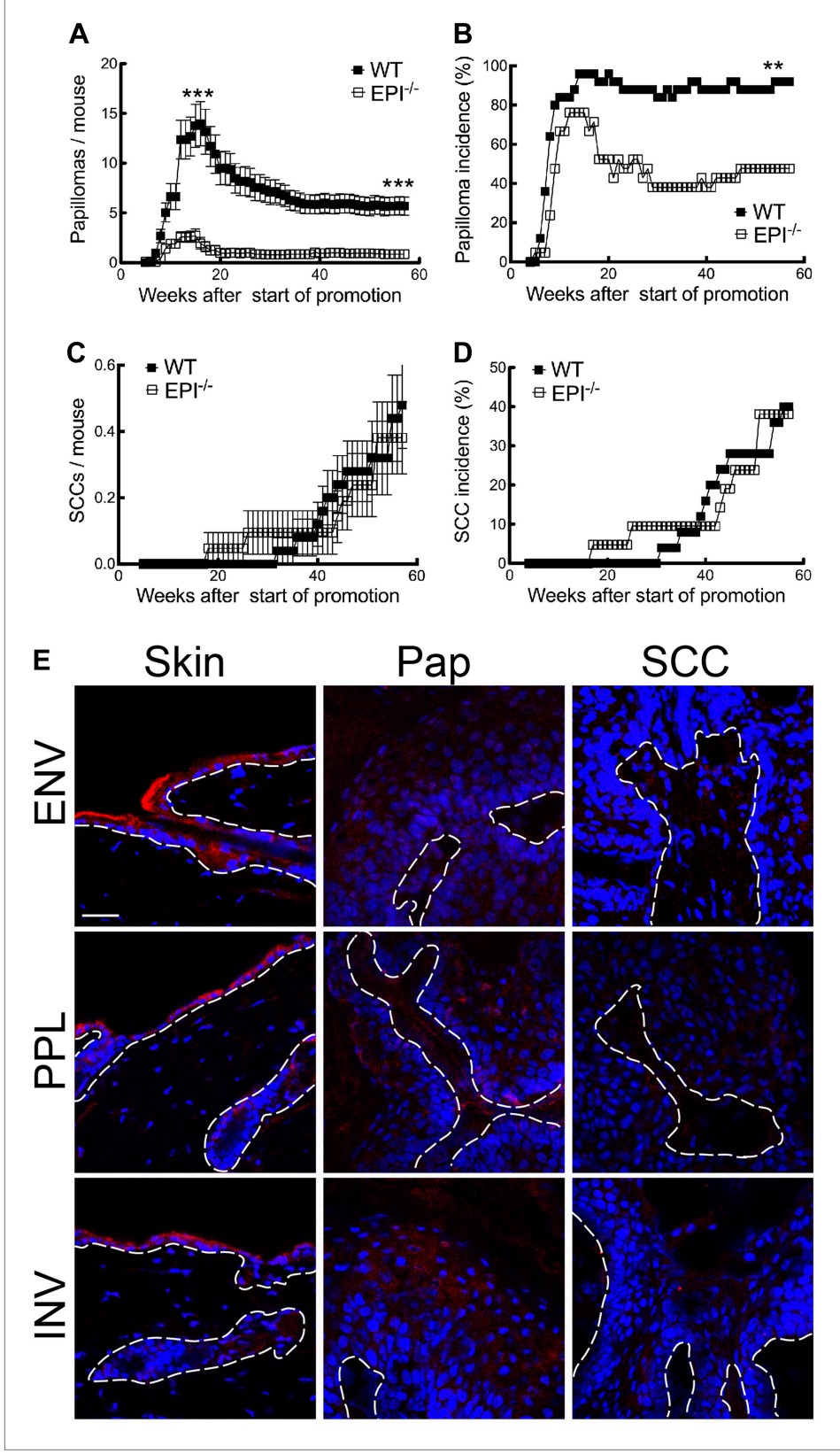

**Figure 1**. Chemical carcinogenesis in EPI−/− and WT mice. (**A**) Average number of papillomas per mouse. (**B**) % mice with one or more papilloma. (**C**) Average number of SCCs per mouse. (**D**) % mice with one or more SCC.
*Figure 1. Continued on next page*

*Figure 1. Continued*

(**A** and **C**) Data are means ± SEM. (**E**) Back skin, papillomas (pap) and SCCs from WT mice were immunostained for Envoplakin, Periplakin, or Involucrin (red) and DAPI (blue). Dotted line indicates basement membrane. Scale bars: 100 µm.

received three topical applications of TPA (or vehicle control), which corresponds to 1 week of promotion, and skin was collected 48 hr after the last treatment.

EPI−/− mice exhibited an exacerbated response to TPA (*Figure 3A*). Their skin was reddened, dry and scaly, whereas that of WT mice was not (data not shown). Histological analysis revealed increased hyperkeratosis (thickened stratum corneum) and parakeratosis (retention of nuclei in cornified cells) (*Figure 3B,C*), consistent with defective desquamation (*Sevilla et al., 2007*). Widening of intercellular spaces in the epidermal basal layer (spongiosis) was extensive in EPI−/− but not WT skin (*Figure 3A*). Epidermal thickness, measured as the distance from the basal to the upper granular layer, was greater in EPI−/− mice treated with TPA or DMBA/TPA than in WT mice, and this correlated with a greater number of suprabasal layers (*Figure 3A,D*). The numbers of keratinocytes with dsDNA breaks and active caspase-3 were similar in EPI−/− and WT epidermis (*Figure 3F,G*).

The number of mitotically active basal cells, measured by PhosphoHistone3 staining, was similar in TPA-treated EPI−/− and WT epidermis, whether or not the skin was pre-treated with DMBA (*Figure 3E*). To examine whether the transit time through the suprabasal layers was altered, mice were injected with BrdU 24hr prior to sacrifice and the position of labelled cells was quantified. The numbers of BrdU positive cells in the basal layer and outermost granular layers of EPI−/− and WT epidermis were not significantly different (*Figure 3H*). Since EPI−/− epidermis contained more suprabasal layers, the transit time of keratinocytes through the epidermis was faster in EPI−/− than WT epidermis. These observations are consistent with a model whereby precocious differentiation can act as a tumour suppressive mechanism in skin (*Guinea-Viniegra et al., 2012*).

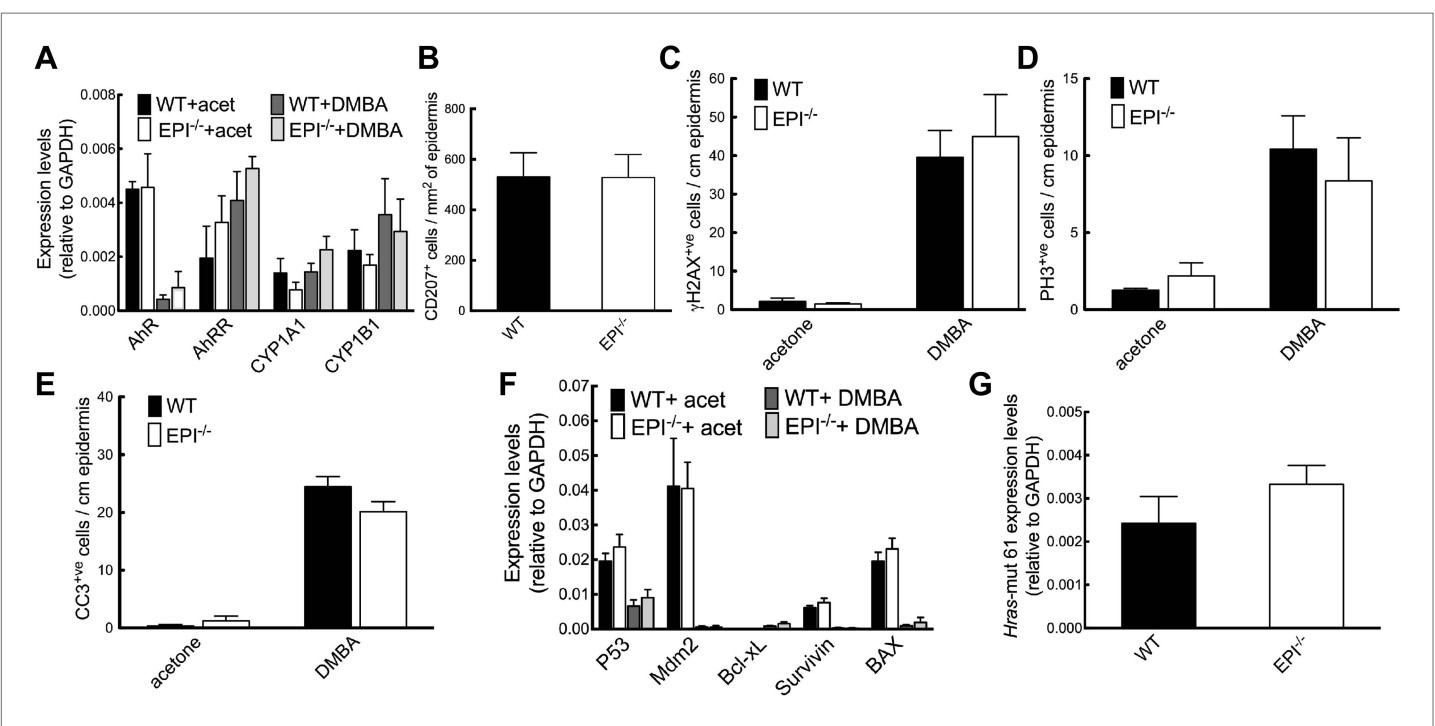

**Figure 2**. Response of EPI−/− and WT mice to DMBA. (**A**) Q-PCR of enzymes responsible for DMBA uptake and metabolic activation. (**B**) Number of CD207+ Langerhans cells per mm² ear epidermis. (**C–E**) Number of epidermal cells that scored positive for γH2AX (**C**), PH3 (**D**) or active caspase-3 (**E**) per cm. 7-cm skin was analysed per marker. (**F**) Q-PCR of anti- and pro-apoptotic genes. (**A–F**) Data in all histograms are means ± SEM of at least three mice per genotype. (**G**) Hras codon 61 mutations in 4 SCCs per genotype were quantified by Q-PCR of genomic DNA.

We conclude that, in contrast to the DMBA response, the response of epidermal keratinocytes to TPA was markedly different in EPI−/− and WT mice.

## EPI−/− mice exhibit an exacerbated inflammatory response to TPA

Because inflammation is a crucial factor in skin tumour development (*Johansson et al., 2008*; *Zamarron and Chen, 2011*) and EPI−/− skin has an abnormal number of DETCs and CD4+CD3+ lymphocytes (*Sevilla et al., 2007*), we examined whether the inflammatory response to short term TPA treatment was altered in skin with a defective epidermal barrier. Consistent with our previous observations (*Sevilla et al., 2007*), vehicle-treated EPI−/− skin had a reduced number of epidermal γδTCR+CD3+ lymphocytes (DETCs) (*Figure 4A*) and EPI−/− γδ T cells exhibited a more rounded morphology, indicative of activation (*Figure 4I*; *Strid et al., 2008*). On TPA treatment, the number of DETCs in both EPI−/− and WT epidermis decreased (*Figure 4A*). In contrast, TPA treatment stimulated recruitment of CD4+CD3+ lymphocytes into the skin of EPI−/− mice to a greater extent than in WT mice (*Figure 4B,J*). On TPA treatment, CD4+ T cells infiltrated EPI−/− but not WT epidermis (*Figure 4J*). Infiltration of granulocytes of the neutrophil family (Cd11b+LY6G+) is a typical sign of chronic skin inflammation, where they accumulate in pustules (*Figure 4D,E*). The number of pustules was significantly higher in TPA-treated EPI−/− compared with WT skin (*Figure 4E*).

TPA treated EPI−/− skin showed a pronounced infiltration of dermal mast cells and eosinophils (*Figure 4F,G*), but not macrophages (F4/80+Cd11b+ cells) (*Figure 4H*). Similar differences in immune cell recruitment in EPI−/− and WT skin were observed when mice were treated with DMBA prior to TPA (data not shown).

Pustule formation and dermal infiltration of mast cells, eosinophils and CD4+ T cells are hallmarks of the acute phase of atopic dermatitis in humans. Thus the response of EPI−/− skin to TPA resembled the acute phase of the human disease and involved recruitment of multiple leukocyte subsets.

## EPI−/− keratinocytes express elevated levels of cytokines and chemokines and an exacerbated lymphoid stress surveillance response

Keratinocytes communicate with different immune cell populations by production of an array of cytokines and chemokines, collectively known as the 'epimmunome' (*Swamy et al., 2010*). Tissue damage triggers upregulation of keratinocyte 'stress-associated' genes such as Rae-1 (*Gasser et al., 2005*) and H60 (*Whang et al., 2009*). These stress antigens engage the immunoreceptor NKG2D expressed by cells of the innate (NK cells and DETC) and adaptive (activated CD8+ T cells, NKT cells) immune system, triggering a lymphoid stress-surveillance response (*Hayday, 2009*).

To investigate how a defective epidermal barrier could lead to an enhanced inflammatory response to TPA, we compared cytokine and chemokine expression in epidermis and dermis of WT and EPI−/− mice. We examined markers of innate, type 1 (cellular), type 2 (humoral) and type 17 (anti-microbial) immunity. In EPI−/− epidermis treated with acetone, transcripts related to innate immunity (in particular RAGE, S100A8, S100A9 and IL-1β) were higher than in WT, and these differences were maintained when transcript levels were upregulated by TPA treatment (*Figure 5A*). Type 1 cytokine transcripts were expressed at similar levels in WT and EPI−/− epidermis and dermis and were not upregulated to the same extent as other types of cytokines by TPA (*Figure 5A*). There was a striking upregulation of type 2 cytokines, including IL-4, IL-6, IL-13, IL-33 and thymic stromal lymphopoietin (TSLP), in epidermis and dermis of EPI−/− TPA-treated mice; this was not observed to a similar extent in WT skin. Type 17 transcripts were upregulated by TPA in epidermis of both genotypes, but were higher in EPI−/− epidermis (*Figure 5A*). Increased transcript levels of a number of chemokines, in particular Cxcl-5 and GM-CSF, were found in TPA treated EPI−/− vs WT skin, in agreement with the increased dermal infiltration of neutrophils and eosinophils. We conclude that the immune response to TPA in EPI−/− skin correlated with expression of pro-inflammatory genes with a type 2/type 17 profile.

Consistent with the type 2 response (*Figure 5A*) and mast cell influx (*Figure 4F*), serum IgE levels were 3.4-fold higher in EPI−/− than WT mice (*Figure 5B*). There was also a higher level of TSLP in EPI−/− serum (*Figure 5C*). TSLP is mainly produced by keratinocytes and is associated with increased resistance to skin carcinogenesis (*Kuramoto et al., 2002*; *Aberg et al., 2008*; *Demehri et al., 2008*, *2012*; *Di Piazza et al., 2012*). TSLP levels are elevated when Notch signalling in skin is impaired (*Demehri et al., 2008*; *Dumortier et al., 2010*). However, EPI−/− and WT epidermis expressed comparable levels of Notch1, 2 and 3, Notch ligand Jag1 and Notch target genes Hey-1, Hes-1 and Hes-5, with or without TPA treatment (*Figure 5D*). Thus, upregulation of TSLP is associated with barrier

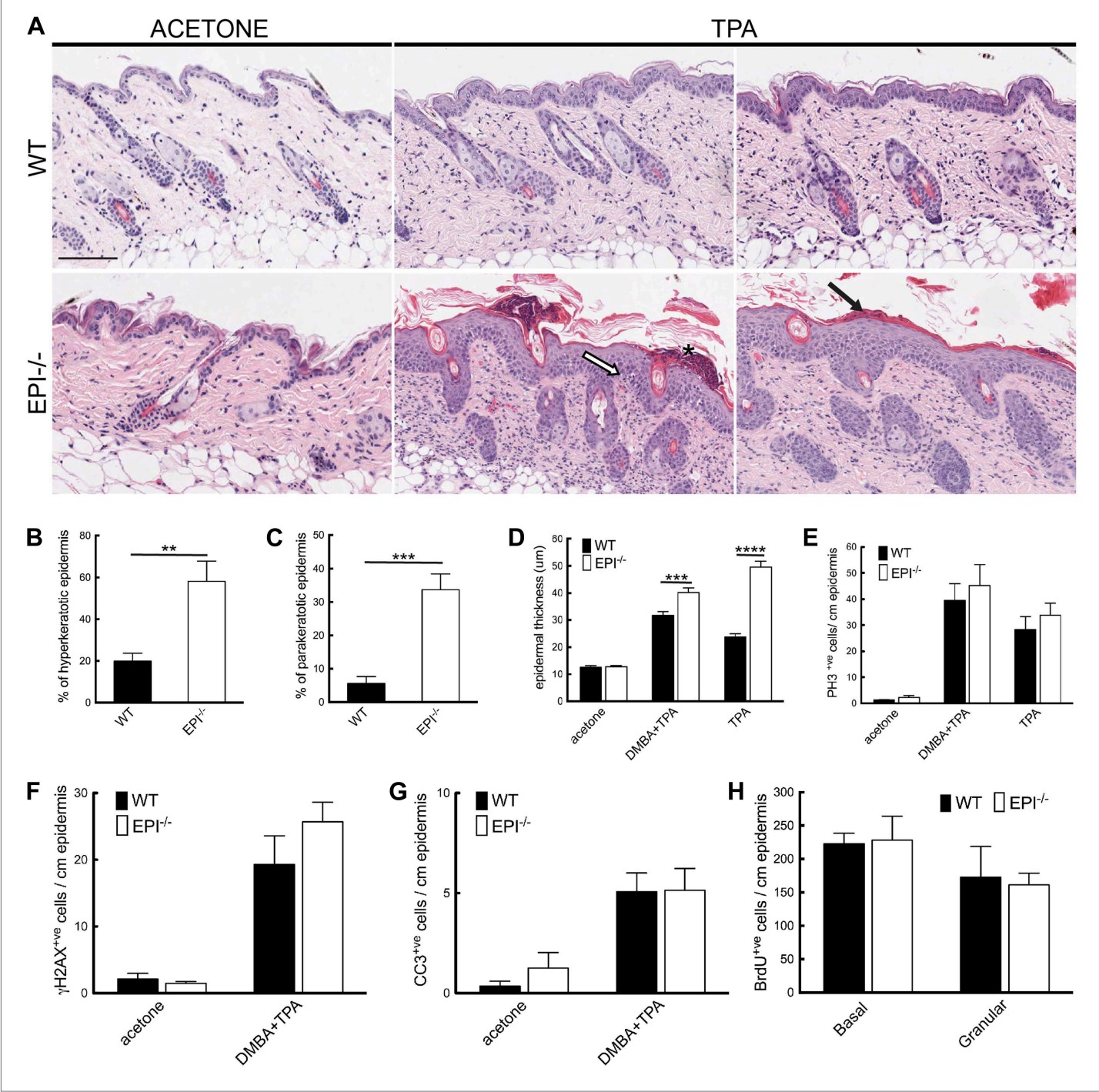

**Figure 3**. Keratinocyte responses to TPA treatment. (**A**) H&E stained skin sections of mice treated three times with acetone or TPA. Asterisk: neutrophil containing pustule; white arrow: spongiosis; black arrow: parakeratosis. % hyperkeratotic (**B**) and parakeratotic (**C**) stratum corneum (8 cm skin analysed per condition). (**D**) Epidermal thickness in μm (8 cm skin analysed per condition). (**E–G**) Number of epidermal cells positively labeled for PH3 (**E**), γH2AX (**F**) and active caspase-3 (**G**) per cm skin (7 cm skin analysed per condition). (**H**) Number of BrdU positive cells in basal and uppermost two granular layers per cm epidermis of mice treated three times with TPA and injected with BrdU 24 hr before harvesting. Data from all graphs represent means ± SEM from at least four mice per genotype. Scale bar: 100 μm.

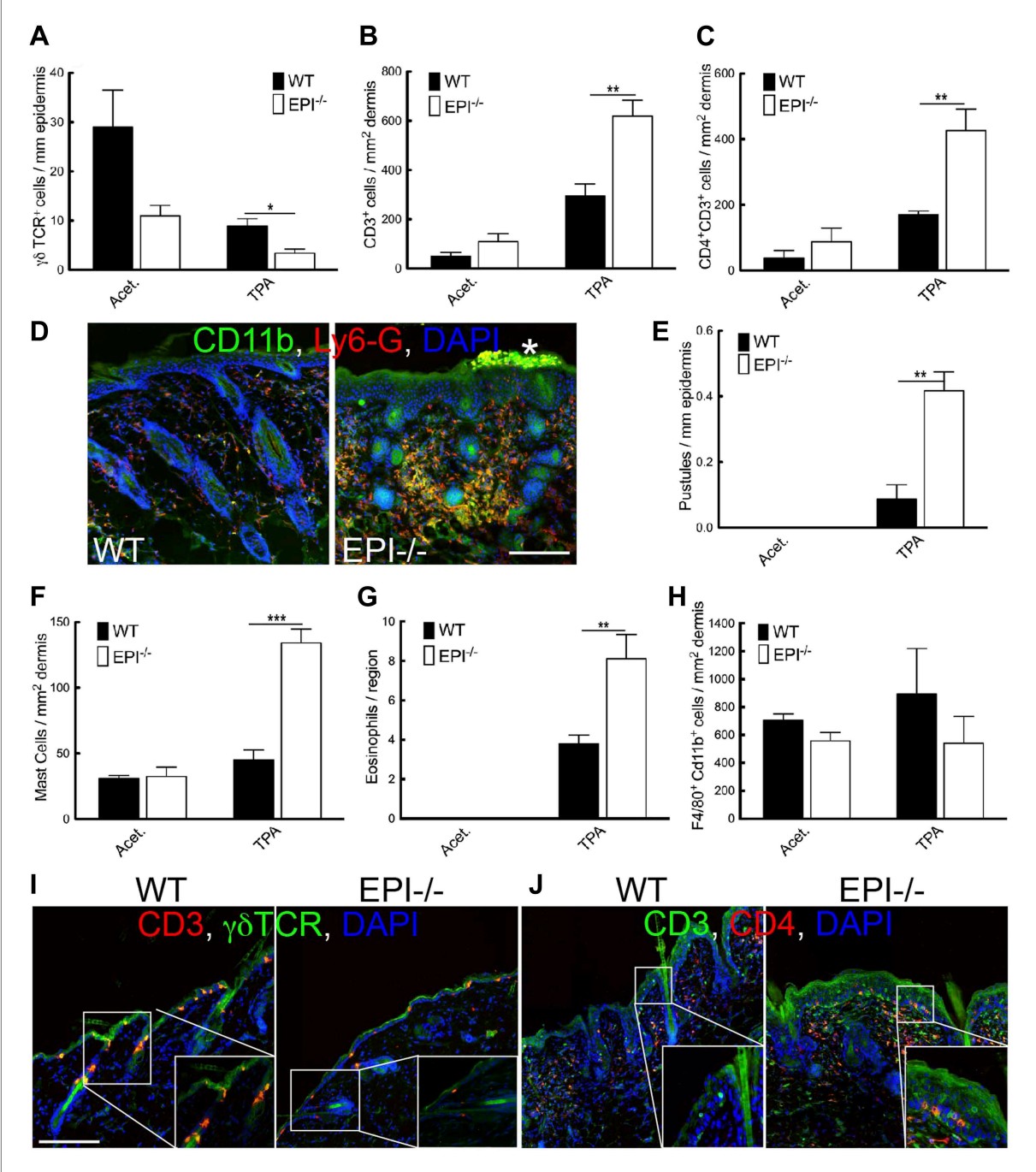

**Figure 4**. Immune cell responses to TPA treatment. (**A–C** and **E–H**) Quantitation of specific T lymphocyte populations (**A–C**), pustules (**E**), mast cells (**F**), eosinophils (**G**), and macrophages (**H**) per mm epidermis or mm² dermis. Data in all histograms are means ± SEM of at least three mice per genotype. (**D**, **I**, **J**) Skin sections of mice treated three times with acetone (**I**) or TPA (**D** and **J**) and labelled for CD11b (green, **D**), Ly6G (red, **D**), CD3 (red **I**, green **J**), γδTCR (green **I**) or CD4 (red, **J**) with DAPI nuclear counterstain (blue). White asterisk: neutrophil pustule. Insets in **I** and **J** are higher magnification views of boxed areas. Scale bars: 100 μm.

defects rather than being a direct consequence of altered Notch signalling (*Demehri et al., 2008*; *Dumortier et al., 2010*).

These results suggested that lymphoid stress-surveillance of tissue damage could underlie the exacerbated inflammatory response to TPA exposure in EPI−/− mice. We therefore measured expression of the keratinocyte 'stress-associated' genes Rae-1 and H60 (*Figure 5E,F*). Rae-1 and H60 levels

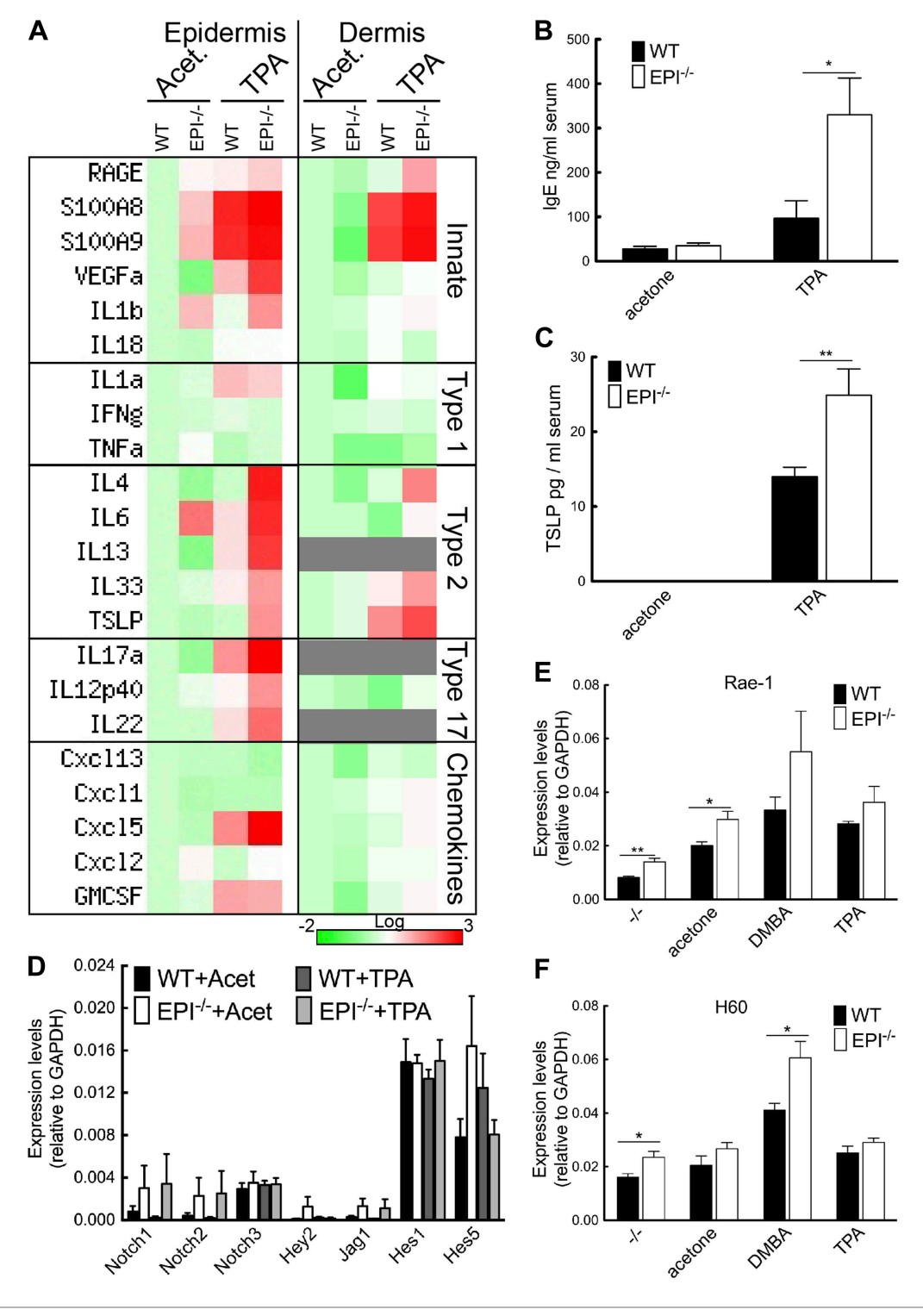

**Figure 5**. Stress signals, cytokine, and chemokine production in EPI−/− and WT skin. (**A**) Heatmap of mRNA levels relative to GAPDH from epidermis and dermis. Each value represents the mean of data obtained from four mice. (**B** and **C**) Serum levels of IgE (**B**) and TSLP (**C**) determined by ELISA. (**D–F**) Q-PCR of indicated mRNAs in epidermis. All histograms represent mean ± SEM (N = 5 untreated, N = 7 acetone, N = 4 DMBA, N = 10 TPA treated mice per genotype).

were significantly higher in EPI−/− than WT epidermis under steady-state conditions. Both were strongly upregulated 24 hr after DMBA treatment, H60 to a greater extent in EPI−/− mice. The effects of TPA treatment on Rae-1 and H60 expression were less pronounced. We conclude that the defective barrier of EPI−/− mice results in elevated expression of keratinocyte stress-associated genes that could prime them for an exacerbated atopic response to TPA.

## The TPA response of EPI−/− skin depends on an immune infiltrate, but is not specific for neutrophils or CD4+ T cells

To investigate the mechanism by which TPA provoked an exacerbated atopic response in EPI−/− epidermis, we used several different inhibitory strategies. Treatment with the anti-inflammatory drug dexamethasone (DXM) markedly reduced epidermal thickening and dermal cellularity (*Figure 6A,B*). DXM also prevented the selective increase in spleen size observed in TPA-treated EPI−/− mice (*Figure 6F*).

Since tumour resistance of Notch deficient skin is attributable mainly to CD4+ T cells (*Demehri et al., 2012*), we examined the effect of intraperitoneal injections of blocking CD4 antibodies. In vivo CD4 depletion was confirmed by quantifying CD4+ cells in the spleen (data not shown). Specific CD4 targeting resulted in augmented skin hyperplasia and dermal inflammation in TPA-treated WT mice, but had no effect on EPI−/− skin (*Figure 6C*). The immunosuppressive role of CD4+ cells in WT skin may be attributable to the regulatory subset of CD4+ cells (CD4+CD25+Foxp3+) (*Teige et al., 2009*).

There was a marked increase in neutrophil infiltration in EPI−/− TPA-treated skin compared to WT (*Figure 4D,E*). However, when circulating neutrophils were depleted by anti-LY6G injection (*Figure 6G*), there was no effect on epidermal hyperplasia and dermal inflammation (*Figure 6D*). We also targeted IL-4 to evaluate whether inhibition of a generalised type 2 immune response could normalize the effects of TPA in EPI−/− skin (*Figure 6E*). We confirmed a decrease in IL-4 serum levels by ELISA (*Figure 6H*). Inhibition of IL-4 reduced the number of CD4+ cells in the dermis (*Figure 6J*), as expected, but had no effect on epidermal thickness (*Figure 6I*) nor on the number of mast cells, eosinophils or epidermal γδ T cells (data not shown).

Taken together, these observations indicate that the exacerbated TPA response in EPI−/− skin involves multiple immune cell types and cannot be normalized by inhibiting one specific leukocyte subset.

## TSLP and NKG2D inhibition suppress the exacerbated TPA response of EPI−/− mice

To analyse the role of the Rae-1/NKG2D/TSLP axis in the exacerbated atopic response of EPI−/− skin to TPA, we interfered with the epidermal lymphoid stress surveillance response by in vivo antibody blockade of TSLP or NKG2D (*Figure 7A*). We confirmed effective inhibition of TSLP by ELISA (*Figure 7B*).

Neutralization of TSLP completely abrogated the atopic dermatitis phenotype in TPA treated EPI−/− mice (*Figure 7A*). Anti-NKG2D also reduced the atopic phenotype, albeit to a lesser extent (*Figures 7A and 8*). TPA-induced epidermal thickening was significantly reduced in TSLP or NKG2D suppressive conditions, without any differential effect on the number of PH3+ basal layer cells (*Figure 7A,C, D*). TSLP treatment reduced the transit time of BrdU + cells from the basal to the outermost granular layers since although the epidermis was thinner the proportion of labelled granular cells was unaffected (*Figure 7E,F*). There was a slight reduction in hyperkeratosis, and parakeratosis was clearly decreased (*Figure 7G,H*). Targeting the Rae-1/NKG2D/TSLP axis also reduced the number of dermal CD4+CD3+ lymphocytes, mast cells, eosinophils, and epidermal pustules in EPI−/− mice (*Figure 8A–D*). Furthermore, under TSLP suppressive conditions the number of epidermal DETCs was increased in EPI−/− skin (*Figure 8E*). Macrophage numbers were unaffected by TSLP or NKG2D blockade (data not shown).

To examine whether the anti-inflammatory effects of blocking TSLP were mediated by down-regulating epidermal expression of inflammatory mediators (*Figure 5A*), we measured expression of representative type 1 and type 2 cytokines. TPA treatment induced a marked and significant release of type 2 cytokines (IL-4, IL-6) in the skin and serum of EPI−/− mice, and this was diminished by systemic administration of anti-TSLP antibodies (*Figure 8F,G*). In contrast, there was no effect on the type 1 cytokines, TNFα and IFNγ. Thus, the inflammation-suppressive effects of systemic TSLP inhibition were mediated at the level of both local and systemic suppression of inflammatory mediator production.

EPI−/− SCCs expressed higher TSLP levels than WT SCC (*Figure 8I*), whereas there were no significant differences in IL-4 and IL-17 (*Figure 8H*). In addition, EPI−/− mice bearing papillomas larger than

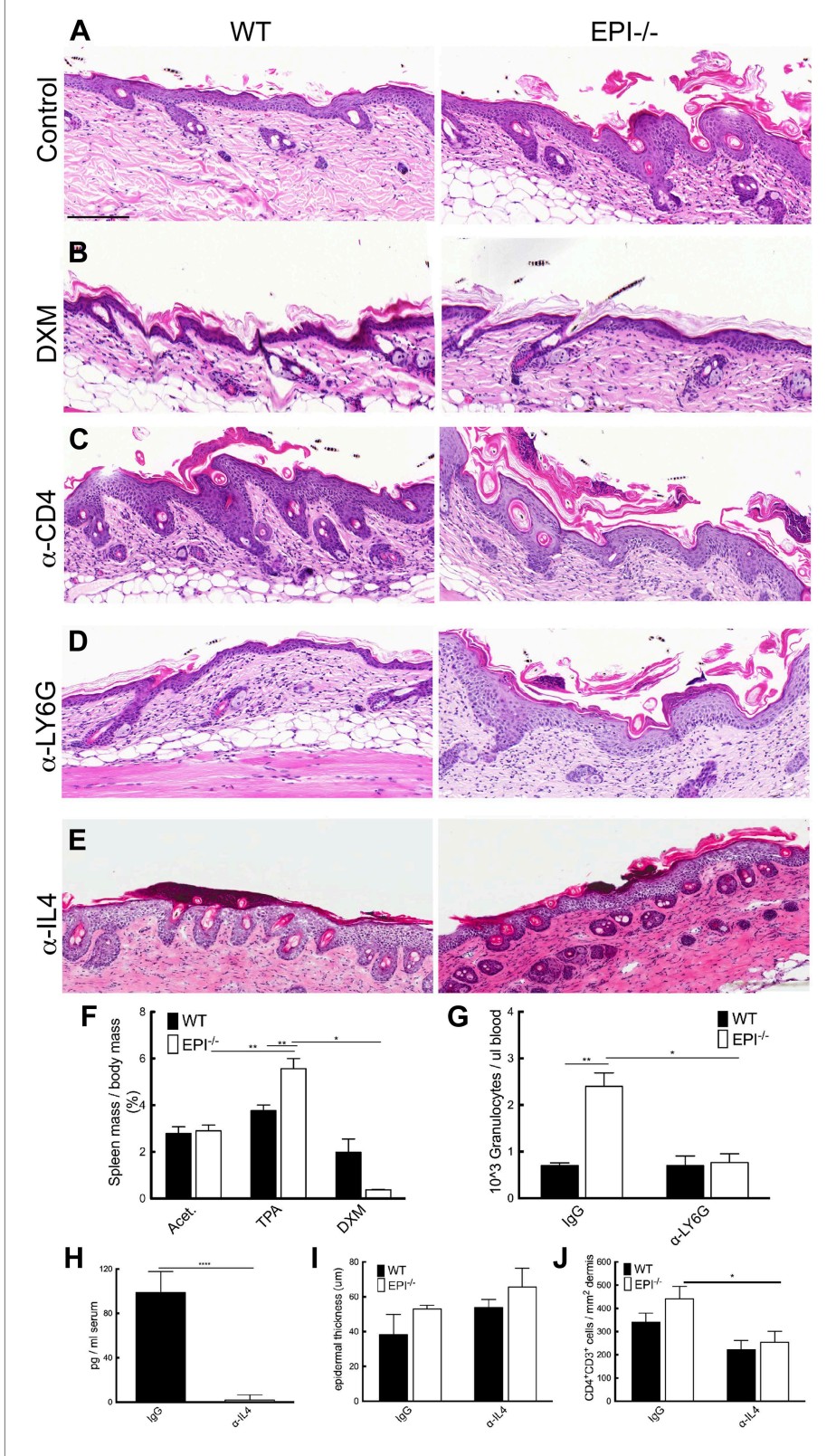

**Figure 6**. Blocking strategies to revert the atopic response to TPA treatment. (**A**–**E**) H&E stained skin sections of WT and EPI−/− mice painted with TPA and injected with IgG (**A**), Dexamethasone (**B**), α-CD4 antibody (**C**), α-Ly6G antibody (**D**) or α-IL4 (**E**). Scale bars: 100 μm. (**F**) Spleen mass of mice treated with acetone, TPA or

*Figure 6. Continued on next page*

*Figure 6. Continued*

TPA + Dexamethasone as % of body weight. Data are means ± SEM from 12 (acetone, TPA) or 3 (TPA + DXM) mice per genotype. (**G**) Numbers of granulocytes per µl blood in TPA treated mice injected with IgG or α-Ly6G antibody. (**H**) Quantification of serum levels of IL-4 in EPI−/− mice treated with IL-4 or control antibodies. (**I** and **J**) Effects of anti-IL-4 on epidermal thickness (µm) (8 cm skin analysed per condition) (**I**) and dermal CD4 + T cells (**J**).

2 mm$^2$ had higher levels of systemic TSLP than EPI−/− mice with smaller papillomas (***Figure 8J***). This indicates that when tumours do arise in an atopic environment, TSLP may have a tumour growth-promoting role. This is in contrast to the reported ability of TSLP to shrink tumours in Notch-deficient skin (***Demehri et al., 2012***), but in agreement with the potent tumorigenic effect of TSLP in breast and pancreatic cancers (***De Monte et al., 2011***; ***Pedroza-Gonzalez et al., 2011***).

We conclude that targeting the Rae1/NKG2D/TSLP axis reverts the atopic phenotype observed in EPI−/− mice upon TPA treatment by interfering with the crosstalk between keratinocytes and immune cells.

## Discussion

Mice deficient in *Evpl, Ppl* and *Ivl* exhibited a striking resistance to benign tumour formation. This was not due to a difference in DMBA-induced initiation, indicating that the tumour-protective effect was not mediated by cell-autonomous changes associated with acquisition of oncogenic mutations. Rather, short-term TPA treatment exacerbated atopic dermatitis in EPI−/− mice, resulting in a phenotype that shared key hallmarks of the acute phase of the disease in humans (***Bieber, 2010***) (***Figure 9***). These hallmarks included epidermal acanthosis, spongiosis, hyperkeratosis, and parakeratosis, systemic type 2 cell expansion with a dramatic accumulation of CD4$^+$ T cells, neutrophils, eosinophils and dermal mast cells, and elevation of serum IgE and TSLP levels.

The enhanced thickening of TPA-treated EPI−/− epidermis reflected an increase in the number of suprabasal, differentiated cell layers and defective desquamation. Proliferation in the basal layer was similar to wild type, but the transit time of cells through the epidermis was faster. These changes in differentiation have previously been shown to be tumour-suppressive in oncogene-driven skin carcinogenesis (***Guinea-Viniegra et al., 2012***), consistent with our observation that EPI−/− mice are resistant to developing papillomas. Upon TPA treatment EPI−/− defective keratinocytes released high levels of chemokines and type 2-type 17 cytokines, including TSLP. This correlated with recruitment of innate immune cells, including mast cells, granulocytes (both neutrophils and eosinophils) and adaptive immune cells, such as CD4$^+$ T cells, which could potentially play a role in immuno-editing of nascent papillomas.

Defective epidermal differentiation in concert with activation of innate and adaptive immune responses is known to create the first line of defence against physical, chemical, and toxic assaults in the skin (***Milstone, 2004***; ***Sherman et al., 2008***). Key players in the crosstalk between keratinocytes and leukocytes are DETCs, since they orchestrate the inflammatory response to pathogen-derived and environmental signals (***King et al., 1999***; ***Moore et al., 2000***). DETCs are already in an activated state in untreated EPI−/− epidermis, as demonstrated by their round shape and reduced number (***Sevilla et al., 2007***). This is most likely a consequence of increased expression of keratinocyte stress signals (Rae-1 and H60), which would be responsible for establishing the observed Th2 immune-surveillance response. We propose that this pre-activated, 'alarmed' condition enables EPI−/− skin to react more promptly and strongly to TPA treatment than WT skin. In contrast to EPI−/− mice, in mice that overexpress activin βA in the epidermis downregulation of DETCs only occurs after DMBA/TPA treatment and is linked to increased skin carcinogenesis (***Antsiferova et al., 2011***). The exaggerated TPA response in EPI−/− mice resulted in increased numbers of CD4+ T cells, neutrophils, mast cells, and eosinophils, and many of these leukocyte populations have been shown to be able to exert tumour protective effects, which could potentially bypass the function of γδ T cells.

Targeting the Rae-1/NKG2D/TSLP axis, by blocking TSLP or NKG2D, suppressed the crosstalk between keratinocytes and immune cells and abrogated the atopic phenotype of TPA-treated EPI−/− skin, as seen by normalisation of epidermal differentiation and inflammation. In mice, epidermal overexpression of TSLP results in an atopic dermatitis phenotype characterized by epidermal hyperproliferation, acanthosis, spongiosis, and hyperkeratosis, as well as mast cell infiltration in the dermis (***Yoo et al., 2005***). In humans, TSLP is implicated in the pathogenesis of both atopic dermatitis and

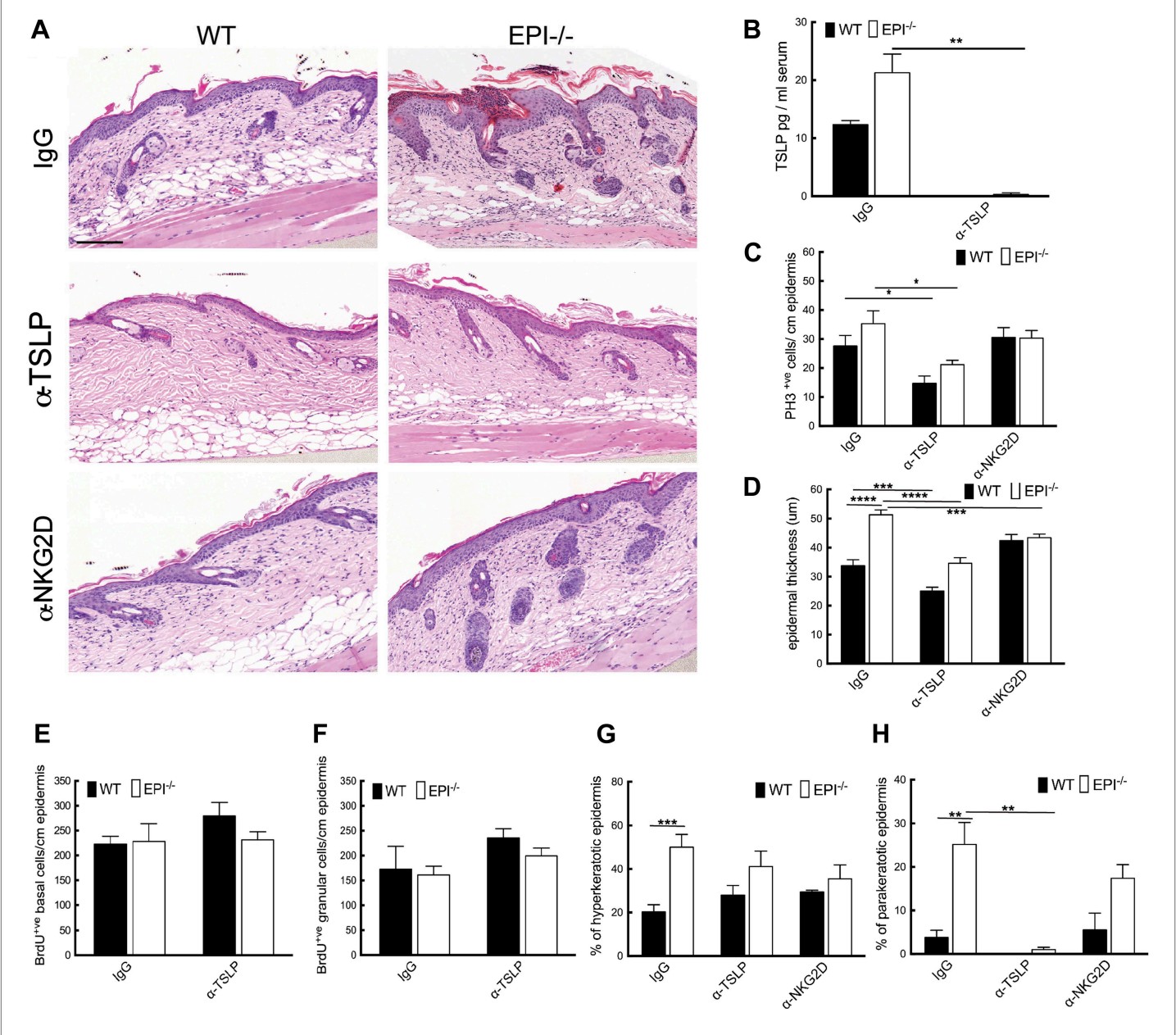

**Figure 7**. TSLP and NKG2D inhibition reduce epidermal responses to TPA. (**A**) H&E stained sections of skin from TPA-treated mice injected with IgG, anti-TSLP, or anti-NKG2D antibodies. (**B**) Quantification of serum levels of TSLP in WT and EPI−/− mice treated with TSLP or control antibodies. (**C** and **D**) Quantification of PH3+ cells (**C**) and epidermal thickness (μm) (**D**). (**E** and **F**) Number of BrdU positive cells in basal (**E**) and uppermost two granular (**F**) epidermal layers in mice treated with TPA and the indicated antibodies. (**G** and **H**) % hyperkeratotic (**G**) and parakeratotic (**H**) stratum corneum. (**C–D** and **E–F**) 7-cm skin analysed per condition.

asthma (*Leonard, 2002*; *Al-Shami et al., 2005*; *Zhou et al., 2005*; *Huston and Liu, 2006*; *Liu, 2006*). TSLP has potent anti-tumour activity in skin carcinogenesis (*Demehri et al., 2012*; *Di Piazza et al., 2012*). Consistent with this, TPA-treated EPI−/− skin produced higher levels of TSLP than WT skin, and was protected from chemically induced papilloma formation.

NKG2D ligands have been shown to be upregulated in many inflammatory lesions and can act as an axis of immunoregulation (*Strid et al., 2008*). Our findings demonstrate that the proposed role for Rae-1 and NKG2D in establishment of the lymphoid stress-surveillance response and consequent atopic phenotype also applies to the situation in which there is a defective epidermal barrier.

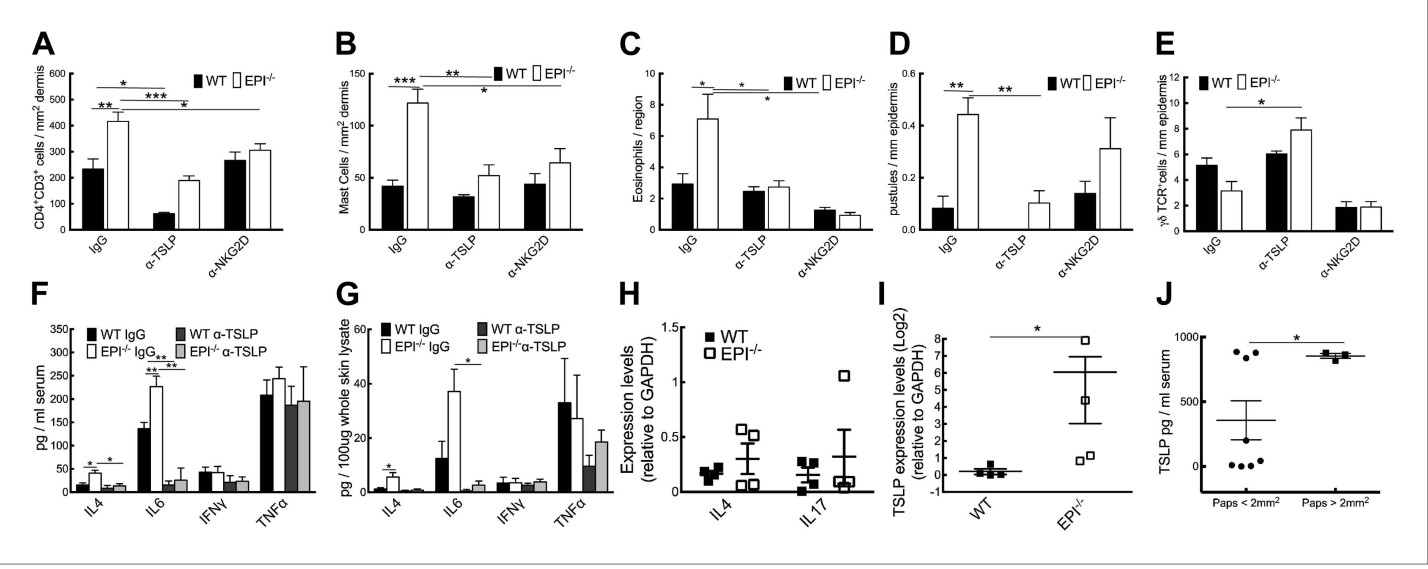

**Figure 8**. Effects of TSLP and NKG2D inhibition and quantitation of cytokine levels in SCCs. (**A**–**E**) CD4$^+$ T cells (**A**), mast cells (**B**), eosinophils (**C**), pustules (**D**), and γδ T cells (**E**) per mm epidermis or mm$^2$ dermis. Data are means ± SEM from at least three mice per genotype. (**F** and **G**) Serum (**F**) and whole skin (**G**) protein levels of the cytokines indicated. Data are means ± SEM from 6 IgG and 3 α-TSLP-treated mice. (**H** and **I**) Q-PCR of mRNAs indicated in SCCs. Data are means ± SEM of 4 SCCs per genotype. (**J**) Serum levels of TSLP in EPI−/− mice bearing papillomas smaller than 2 mm$^2$ or at least one papilloma larger than 2 mm$^2$. Data are means ± SEM of at least three mice per group.

Although two-stage chemical skin carcinogenesis has been used extensively to elucidate the role of individual components of the immune system (specific cell types, cytokines and signalling molecules), ours is the first mouse model to link the lack of structural proteins and consequent defective epidermal barrier to cancer susceptibility. Epidemiological studies of the association between allergic disease and cancer risk have given conflicting results, in part because of the difficulty of factoring in immune suppressive treatments for the disease and in part because the association can be positive or negative according to cancer type (*Arana et al., 2010*; *Van Hemelrijck et al., 2010*; *Wiemels et al., 2011*; *Jensen et al., 2012*; *Kaae et al., 2013*). Our study provides a mechanistic framework for further analysis of the link between atopic dermatitis and cancer incidence.

## Materials and methods

### EPI−/− and control mice

Mice deficient for *envoplakin, periplakin,* and *involucrin* (EPI−/−) were generated as previously described (*Sevilla et al., 2007*) by interbreeding single knockout mice for *envoplakin* (*Maatta et al., 2001*), *involucrin* (*Djian et al., 2000*), and *periplakin* (*Aho et al., 2004*). Single knockout mice were generated and maintained on a variety of genetic backgrounds and as a result there was a potential contribution of Sv129, C57BL/6, BALB/c and nu/nu to EPI−/− animals. Since DMBA/TPA sensitivity is affected by genetic background (*Hennings et al., 1993*; *Woodworth et al., 2004*), DNA of 4 EPI−/− mice (each from a different breeding pair) was subjected to genome scanning by the Jackson Laboratory (Bar Harbor, ME, USA). Single nucleotide polymorphisms (SNPs) were used to determine the relative contributions of different genetic backgrounds. EPI−/− mice were primarily Sv129 (40.98%, ±1.52) and C57Bl/6 (51.39%, ±1.62), with a small contribution of BALB/c (4.7%, ±1.24). The nu/nu mutation was not detected. Based on these results, the F2 generation of crosses between Sv129 and C57Bl/6J mice was used as the WT control (51.82%, ±3.35 Sv129; 49.92%, ±1.98 C57Bl/6) for all experiments.

### Two stage carcinogenesis

Chemical carcinogenesis experiments were performed as previously described (*Abel et al., 2009*). All experiments were carried out under the terms of a UK Government Home Office project licence, with local ethical approval. The back skin of 7-week-old female mice was shaved with electric clippers. 3 days later animals that did not show signs of hair regrowth received a topical application of 100 nmol

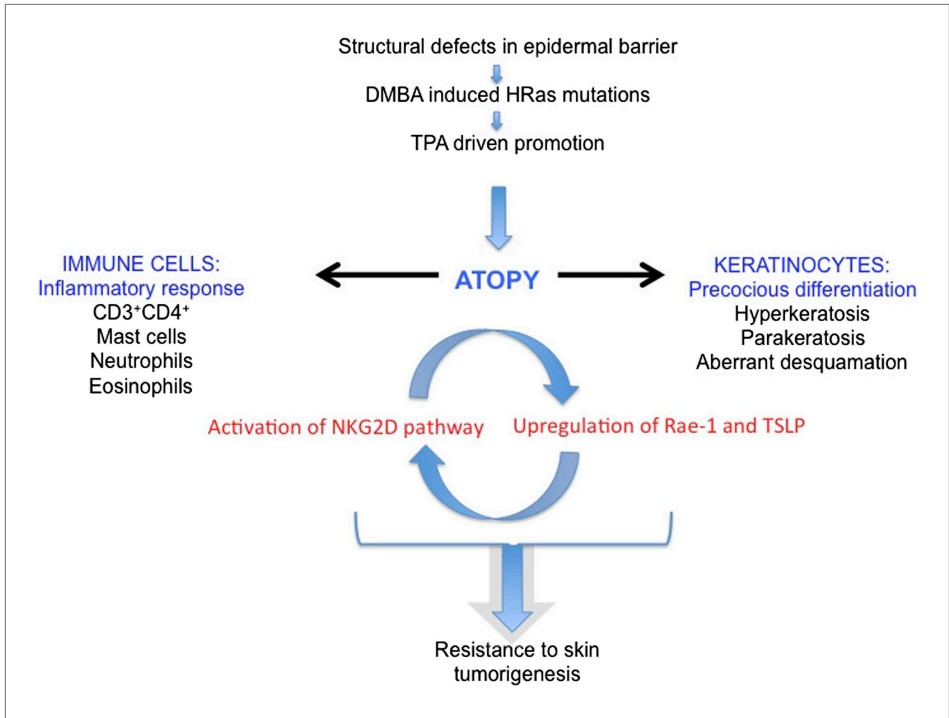

**Figure 9**. Model of the role of an epidermal barrier defect in tumour protection. EPI−/− mice lack three cornified envelope proteins, resulting in a defective epidermal barrier. Topical application of DMBA induces H-Ras mutations, as in wild-type mice. Topical TPA treatment elicits an exaggerated atopic response, characterized by altered keratinocyte differentiation and an enhanced inflammatory response. Epidermal production of TSLP and activation of the ligand-NKG2D pathway on immune cells are proposed to contribute to tumour protection.

(25 μg) DMBA (Sigma–Aldrich) in 200 μl acetone, followed by three times weekly applications of 6 nmol (3.7 μg) of TPA (Sigma-Aldrich, Dorset, UK) in 200 μl acetone for 15 weeks. As controls, mice (5–10 animals/group) were subjected to the same protocol, but substituting DMBA or TPA with acetone. Papillomas and SCCs were recorded once a week for up to 57 weeks after the start of promotion. Recordings were made by an observer who was 'blinded' to the experimental groups. 1 hr before culling, mice were injected intraperitoneally with 50 mg/kg bodyweight 5-bromo-2'-deoxyuridine BrdU (Sigma-Aldrich). In some experiments, age and gender matched mice received 1 application of DMBA and/or 3 applications of TPA on alternating days. Skin was harvested 24 hr after DMBA painting or 48 hr after the last TPA treatment.

## Tissue processing, immunohistochemistry and immunofluorescence labelling

For paraffin sections, tissue was fixed in 10% neutral buffered formalin for 24 hr and embedded in paraffin after dehydration. For frozen sections, tissue was submerged in OCT embedding matrix (Raymond A Lamb, UK) and frozen on dry ice. 4–6 μm (paraffin or frozen) sections were prepared and collected onto glass slides.

Quantitation of the proportion of the epidermis with a parakeratotic or hyperkeratotic stratum corneum was performed on hematoxylin-eosin stained paraffin sections. A minimum epidermal length of 7 cm was analysed from ≥4 mice per condition. Parakeratosis was defined as retention of nuclei in the cornified layers. Hyperkeratosis was defined as abnormal thickening of the stratum corneum relative to wild type, untreated stratum corneum.

BrdU, PH3, and Caspase-3 were detected in formalin-fixed paraffin-embedded sections. Sections were dewaxed and rehydrated on an automated Leica ST5020, then treated for antigen-retrieval and incubated with primary antibody at the indicated dilution. Secondary antibodies were diluted 1:250 and probed using the Bond Intense R Detection kit (Leica Microsystems; Wetzlar, Germany). Sections were scanned and analysed using the Ariol SL-50 system (Applied Imaging Corp., San Jose).

Prior to immunofluorescence staining, frozen sections were air-dried for 10 min at room temperature and fixed in 4% PFA/PBS pH 7.4 for 10 min. Sections were blocked with 2% BSA, 0.02% fish skin gelatin, and 10% goat serum for 1 hr in PBS at room temperature. Sections were incubated in unconjugated or fluorochrome-conjugated primary antibody and DAPI overnight at 4°C, washed in PBS and then mounted using DAKO mounting reagent (DAKO). AlexaFluor 488- or 633-conjugated goat anti-rabbit or anti-mouse IgG (Invitrogen) were diluted 1:1000 and used for detection of unconjugated primary antibodies. The following species-specific antibodies conjugated with Alexafluor 488 or 555 or 647 were used at a dilution of 1:100: anti-CD3 (clone 17.A2; BDPharmingen), anti-CD4 (clone RM4-5; eBioscience), anti-CD8α (clone 53-6.7; BioLegend), anti-γδ TCR (clone GL3; BDPharmingen), anti-F4/80 (clone BM8; eBioscience), anti-Ly6G (clone 1A8; BDPharmingen), anti-CD207 (clone eBioRMUL.2; eBioscience) and anti-CD11b (clone M1/70; eBioscience). In-house produced antibodies against periplakin, envoplakin and involucrin were also used (*Ruhrberg et al., 1996*, *1997*; *Sevilla et al., 2007*). The following species-specific unconjugated antibodies were used at the dilutions stated: anti-PH3 (Upstate rabbit polyclonal, catalogue number 06-570, 1:500), anti-BrdU (Abcam sheep polyclonal, ab1893, 1:500), anti-caspase-3 (Cell Signaling rabbit monoclonal, catalogue number 9664, 1:100) and anti-γH2AX (05-636; 1;10; Millipore0). Mast cells were detected with toluidine blue, and eosinophils were detected by Congo red staining.

Hematoxylin and eosin–stained sections of skin and tumours were graded in a blinded manner by two experts in mouse skin pathology, as described previously (*Owens and Watt, 2001*). When quantifying the number of cells that were positive for a particular marker, at least 6 fields per treated skin or tumour section were analysed.

## Quantitative RT-PCR

RNA was extracted from whole skin, tumours or epidermis that had been scraped from skin following heat treatment for 60 s at 56°C in PBS +10 mM EDTA or from cultured keratinocytes using the RNeasy mini kit (Qiagen, UK). RNA was quantified using a ND-100 NanoDrop spectrophotometer (NanoDrop Technologies). cDNA was prepared using a Superscript III First-Strand Synthesis Supermix for qRT-PCR kit (Invitrogen) according to the manufacturers' instructions.

Quantitative RT-PCR was carried out using designed primers and fast SYBR Green or Taqman probes with Taqman Fast Universal PCR Master Mix (Applied Biosystems) and run on an ABI Prism 7900HT Sequence detection system (Applied Biosystems). The RT-PCR was run for 40 cycles and specificity of the reactions was determined by subsequent melting curve analysis. Quantitation was based on ΔCt calculations using the SDS analysis software and all samples were compared against mouse GAPDH as a house keeping control. Matrix visualization was performed using the online Matrix2PNG tool (*Pavlidis and Noble, 2003*). Primers and TaqMan probes are listed in *Supplementary file 1*.

## TP53 sequencing and mutant specific HRas qPCR

Genomic DNA was isolated from SCCs using the NucleoSpin kit (Macherey–Nagel) as per the manufacturer's protocol. PCR amplification of *HRas* and *p53* exons of interest was performed on 100 ng of DNA using primers described previously (*Niemann et al., 2007*). PCR products were purified and sequenced. To quantify the number of DMBA-induced codon-61 *Hras* CAA->CTA mutations, mutant-specific primers were used as previously described (*Modi et al., 2012*). Obtained Ct values were normalised against genomic GAPDH (gGAPDH).

## Peripheral blood collection, white blood cell analysis, and plasma preparation

Mice were anesthetized with Isoflurane (Anaesthesia, UK) and then blood for differential white blood cell counts was collected by cardiac puncture in the presence of EDTA (1.75 mg of EDTA dihydrate solution per ml of blood) to prevent coagulation. For plasma protein analysis, blood was allowed to clot at room temperature for at least 1 hr. After centrifugation at 14,000 rpm for 15 min at 4°C the serum was collected, aliquoted, and kept at −80°C until analysis.

## Measurement of cytokine production in whole skin and serum

Whole skin lysates and serum were collected, snap-frozen in liquid nitrogen and kept at −80°C prior to analysis. The Cytometric Bead Array Cytokine Flex Kit (BD) was used following the manufacturer's instructions. Samples were run on a LSR-II flow cytometer and the data were analysed with BD

Cytometric Bead Array software. Cytokine production was determined as picograms per ml serum or mg of whole skin lysate.

## Quantification of ear epidermal Langerhans cells

Epidermal sheets were prepared from mouse ears that had been split into dorsal and ventral sides and floated dermal side down for 2 hr at 37°C in 20 mM EDTA. Epidermal sheets were gently lifted from the dermis, washed in PBS and fixed in cold acetone for 20 min at −20°C. After washing in PBS, epidermal sheets were incubated for 1 hr at room temperature with 2% (g/vol) BSA in PBS and stained overnight at 4°C with CD207 Alexa 647 conjugated antibody (eBioscience, clone eBioRMUL.2) and DAPI. After extensive washing, epidermal sheets were mounted on slides with DAKO mounting medium and visualized by confocal microscopy.

## In vivo injection of dexamethasone and inhibitory antibodies

Functional grade purified antibodies against CD4 (16-0042, clone RM4-5 and clone GK1.5, rat IgG2bK; eBioscience), TSLP (MAB555, clone 152614; R&D System, 16-5491, clone eBio28F12; eBioscience ), NKG2D (catalogue number BE0111; BioXCell), LY6G (clone 1A8; BioXCell), IL-4 (catalogue number BE0045; BioXCell), and appropriate isotype controls (BioXCell Rat IgG2a, catalogue number BE0089 for TSLP and Ly6G; Rat IgG2b, catalogue number BE0090 for CD4; hamster IgG catalogue number BE0091 for NKG2D; Rat IgG1, catalogue number BE0088 for IL-4) were injected intraperitoneally at a dose of 0.1 mg/mouse. Dexamethasone 21-phosphate disodium salt (DXM) (D1159; Sigma) was injected at 10 mg/kg mouse body weight, the same volume of PBS being injected in control mice.

## Serum ELISAs

TSLP (Quantikine mouse TSLP kit, R&D Systems), IL-4 (ELISA IL-4 Ready-set-go, eBiosciences), and IgE (Bethyl Laboratory) immunoassays were performed on mouse serum according to the manufacturer's instructions.

## Whole skin lysates

Whole skin protein lysates were prepared by mechanical lysis using ceramic beads (Precellys) in NP-40 buffer (50 mM Tris–HCl, 150 mM NaCl, 1% NP-40, pH.7.4) with complete protease inhibitor cocktail (Roche, Burgess Hill, UK). Tissue was homogenized twice at 1600 rpm for 50 s, then centrifuged for 20 min 200 µl aliquots were snap frozen and kept at −80C.

## Statistics

Statistical analyses were performed using GraphPad Prism (GraphPad Software). The statistical significance of differences between most experimental groups was determined with a two-tailed Student's t-test for unpaired data or Fisher's exact test. For analysis of differences between WT and EPI−/− mouse tumour burden Mann Whitney testing was performed. For analysis of differences between WT and EPI−/− mouse tumour incidence a test of difference in proportions was used. p-values are indicated with: * $0.01 < p < 0.05$, ** $0.01 < p < 0.001$, *** $0.0001 < p < 0.001$, **** $p < 0.0001$.

## Acknowledgements

This work was supported by a FEBS postdoctoral fellowship to SC, a JSPS fellowship to KN, a SRS fellowship to SQ, and funds to FMW from Cancer Research UK, the Wellcome Trust, MRC and the EU FP7 programme (HEALING). We thank Rachida Nachat and Lisa Sevilla for their input and core facilities of the CRUK Cambridge Research Institute for superb technical assistance.

## Additional information

### Competing interests

FMW: Deputy Editor, *eLife*. The other authors declare that no competing interests exist.

### Funding

| Funder | Author |
| --- | --- |
| Medical Research Council | Sara Cipolat, Esther Hoste, Ken Natsuga, Sven R Quist, Fiona M Watt |

| Funder | Author |
|--------|--------|
| Wellcome Trust | Sara Cipolat, Esther Hoste, Ken Natsuga, Sven R Quist, Fiona M Watt |
| European Union | Sara Cipolat, Esther Hoste, Ken Natsuga, Sven R Quist, Fiona M Watt |
| Cancer Research UK | Sara Cipolat, Esther Hoste, Ken Natsuga, Sven R Quist, Fiona M Watt |

The funders had no role in study design, data collection and interpretation, or the decision to submit the work for publication.

## Author contributions

SC, EH, Conception and design, Acquisition of data, Analysis and interpretation of data, Drafting or revising the article; KN, Conception and design, Acquisition of data, Analysis and interpretation of data; SRQ, Acquisition of data, Analysis and interpretation of data; FMW, Conception and design, Analysis and interpretation of data, Drafting or revising the article

## Ethics

Animal experimentation: All experiments were carried out under the terms of a UK Government Home Office project licence (PPL 80/2378), with local ethical approval from King's College London and the University of Cambridge.

# Additional files

### Supplementary file

• Supplementary file 1. Primers and TaqMan probes used.

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
