## [Decision Letter]

Thank you for sending your work entitled “Epidermal barrier defects link atopic dermatitis with altered skin cancer susceptibility” for consideration at *eLife*. Your article has been favorably evaluated by a Senior editor, a Reviewing editor, and 3 reviewers, one of whom, Valerie Horsley, has agreed to reveal her identity.

As you can see, the reviewers' comments are favorable on the whole although there are some issues that will need to be addressed satisfactorily before proceeding with publication in *eLife*. The Reviewing editor and the other reviewers discussed their comments before we reached this decision, and the Reviewing editor has assembled the following comments to help you prepare a revised submission.

The reviewers contrasted your manuscript and its findings with that of the Kopan Cancer Cell paper (2012), both of which explore the relationship of skin barrier and tumor initiation and invoke TSLP. There was general agreement that the Kopan paper was more novel at the time and the reviewers were somewhat split regarding whether the current paper sufficiently extends mechanistic aspects regarding the link between skin barrier and tumorigenesis to a level appropriate for *eLife*. The reviewers did agree that the link between barrier and tumorigenesis is more clearly established in the Watt paper, since the Notch loss of function model in the Kopan paper could alter non-barrier mechanisms to control tumor formation. They also concur that most of the data presented are convincing. However, they also feel that in addition to the provided blocking antibodies experiment, more mechanistic insights are needed to make this into a clear case for *eLife*. Specifically, the most important question becomes how TSLP is acting, since Kopan's lab has implicated TSLP in the tumor formation following barrier issues. Kopan has suggested that TSLP acts on T cells (Dehmehri et al 2012) and a recent paper by Diana Baustista shows nicely that TSLP signals in neurons in the skin (Wilson et al Cell 2013). If the authors can establish whether TSLP acts directly on inflammatory or other cell populations to control tumor formation, this would represent the kind of mechanistic advance requisite for *eLife*. The other comments listed, while extensive, will be relatively straightforward, but this is the only major experimental issue that the authors should address.

The reviewers felt that the link clinically between atopic dermatitis and skin cancer is weak, and here, some of this can be handled textually, by making the paper crisper in its focus and making fewer sweeping statements. In addition, there are some specific comments, which will strengthen the paper, including substantiating the mechanism, and which are lengthy but seem relatively straightforward to address.

*Experimental issues/interpretations*
*of the data:*

1) While EPI-/- mice have defects in skin tumor initiation, it is possible that these genes also play a role in tumor progression. Are Envoplakin, Periplakin or Involucrin expressed in WT skin papillomas or SCCs? If so, they may play additional roles once the skin tumor is already formed.

2) The authors nicely show that the mutagenic effect of DMBA is unaffected by the EPI mutations. The pro-inflammatory effect of TPA, however, is strongly exaggerated in EPI-/- mice. This is likely because barrier defect in these mice makes them more susceptible to the toxic effect of TPA. The authors note that there is increased expression of some inflammatory genes, genes involved in lymphoid stress immune-surveillance (e.g., RAE1) and barrier defense (e.g., TSLP). Blocking TSLP and NKG2D (receptor for RAE1) significantly reduce the inflammatory response. However, another group found that a reduction in DETC number resulted in increased skin tumorigenesis (Antisferova, Nature Communications, 2011). Although EPI-/- mice also have fewer DETCs, they develop fewer tumors. Could the overall increase in immune activation in EPI-/- mice render DETC function dispensable? Some comments on this difference would be helpful.

3) Are there any differences in the presence of alpha/beta T cells in EPI-/- epidermis?

4) The authors use antibodies against cytokines to suppress cytokine signaling during TPA treatment which many of these cytokines (in particular TSLP) have been implicated before. An extension of this mechanistic angle should implicate which cell types are responsible for the suppression of tumorigenesis. How these signals act in the immune cells etc.

5) A model might help this manuscript to better depict mechanistic findings of this manuscript. As presented the paper appears to be a set of fairly standard experiments with histology and RT-PCR.

6) The study reports two sets of findings: first, that EPI mice are protected from DMBA/TPA induced skin papillomas; second, that EPI mice have exaggerated inflammatory response to TPA. It is not clear whether the two are related and whether (and how) the latter explains the former. Normally, this would be tested using either mice deficient in the suspected components (TSLP, NKG2D), or by blocking these proteins with antibodies. The reviewers agreed that neither approach can be reasonably used here because the mice are already triple knockouts and because the treatment with antibodies for several months is not practical.

The results overall fit with the emerging idea (and epidemiological data) that allergic defenses may confer protection against environmental carcinogenes. One possible interpretation of the results is that structural barrier defect caused y EPI mutations results in compensatory immune-mediated barrier defense that confers protection. The mouse model can be very useful to examine the mechanisms of protective effects of immune stress surveillance response.

*Textual issues/interpretations*
*of the data:*

1) The authors may want to consult with an immunologist regarding their terminology and the choice of genes used as indicators of different types of immune responses. For example, IL-12p40 is not a Th17 marker and IL-6 is not a Th2 marker. These cytokines are made by myeloid cells and affect multiple responses; they are not produced by T cells.

2) In addition, the article could benefit from some heavy editing. For example, the authors present as a fact that there is a causal relationship between barrier impairment, structural proteins, and atopic dermatitis. But the authors should use greater precision. In humans, mutations in FLG (a structural protein) are linked to ichthyosis vulgaris and associated with atopic dermatitis and atopic disorders, such as asthma and allergic rhinitis. But the authors state causality, when genetically correlation has been defined. Mouse models of FLG-/- display some features of AD, but not allergic airway inflammation (asthma model). Moreover these findings are more in question now that the animal model (flaky tail) has been shown to be a compound mutation of FLG and Tmem79.

There is true controversy that cancer is associated with atopy. Some references to support this model are Hwang CY (2012) Int J Cancer and Wedgeworth E (2011) Br JDerm but many dermatologists feel that the connection is more tenuous and may be driven by severe atopics being treated with heavy immunosuppressants, which have been associated with increased risk of malignancies. UV light therapy is also a potential treatment that would underlie an increase risk of skin cancer. These are confounding issues for patient data.

3) The extent to which DMBA/TPA treatment is a model for human skin cancer needs to be better addressed in the manuscript as most human SCC do not have H-ras mutations. These issues are mentioned, but typically toward the end of arguments, and need to be addressed before a more simplistic interpretation is depicted as the truth.

4) Authors should read through and edit the entire manuscript. There are lots of circuitous arguments, typographical errors, and information provided in the wrong sections.

*Methods/statistics/clarifications*:

1) How was the function of the antibody blockade of TSLP or NKG2D measured? What percent neutralization was achieved?

2) How was RNA normalized for the acanthosis observed in EPI-/- skin? Basically, if RNA levels are higher, does this reflect an upregulation of the gene or simply that there are more suprabasal cells in the skin? Was RT-PCR deltaCt normalized to B2u or some other ubiquitous gene?

3) In Figure 3, how is the percent of hyperkeratotic and parakeratotic epidermis quantified?

4) Are wt animals +/+;+/+;+/+ or compound heterozygotes. Nu/nu is not a genetic background but rather exists on multiple genetic backgrounds. This is confusing in the Methods section.

5) How representative are the 4 EPI-/- mice of genetic variation?

---

## [Author Response]

*The reviewers contrasted your manuscript and its findings with that of the Kopan Cancer Cell paper (2012), both of which explore the relationship of skin barrier and tumor initiation and invoke TSLP. There was general agreement that the Kopan paper was more novel at the time and the reviewers were somewhat split regarding whether the current paper sufficiently extends mechanistic aspects regarding the link between skin barrier and tumorigenesis to a level appropriate for eLife. The reviewers did agree that the link between barrier and tumorigenesis is more clearly established in the Watt paper, since the Notch loss of function model in the Kopan paper could alter non-barrier mechanisms to control tumor formation. They also concur that most of the data presented are convincing. However, they also feel that in addition to the provided blocking antibodies experiment, more mechanistic insights are needed to make this into a clear case for eLife. Specifically, the most important question becomes how TSLP is acting, since Kopan's lab has implicated TSLP in the tumor formation following barrier issues. Kopan has suggested that TSLP acts on T cells (Dehmehri et al 2012) and a recent paper by Diana Baustista shows nicely that TSLP signals in neurons in the skin (Wilson et al Cell 2013). If the authors can establish whether TSLP acts directly on inflammatory or other cell populations to control tumor formation, this would represent the kind of mechanistic advance requisite for* eLife.

As the reviewers point out, our model is distinct from the Kopan model in that tumour resistance is directly attributable to loss of epidermal barrier proteins, rather than to disturbed barrier function as a consequence of deleting Notch receptors.

Kopan and co-workers attributed the anti-tumorigenic action of TSLP in Notch deficient skin mainly to CD4+ T cells. We now demonstrate that injection of CD4 blocking antibodies does not abrogate the atopic phenotype of TPA treated EPI-/- mice (new Figure 6). There is no difference in the abundance of CD8+ cells between TPA treated EPI-/- and WT skin, but there is a marked increase in neutrophil infiltration. We therefore depleted neutrophils with anti-LY6G antibody, and now show no reduction in hyperplasia and dermal inflammation (new Figure 6). Nevertheless, treatment with the broad-spectrum anti-inflammatory drug dexamethasone (DXM) markedly reduces the TPA response in EPI-/- skin (new Figure 6). We conclude that, in contrast to the Kopan model, multiple immune cell types contribute to the tumour resistance of the EPI-/- model.

One of the novel features of our study, again distinct from the Kopan model, is the demonstration that an intrinsic barrier defect results in activation of the Rae-1/NKG2D/TSLP axis. This provokes a lymphoid stress-surveillance response, which underlies the atopic phenotype in TPA treated EPI-/- mice, since there is reduced dermal infiltration of specific immune cell populations and a reduced keratinocyte response when TSLP or NKG2D is blocked (Figure 7; Figure 8).

Kopan and collaborators show that when mice are treated with calcipotriol, a potent TSLP inducer, tumors shrink over time. In contrast, an interesting feature of our model is that when papillomas do arise in EPI-/- mice they have a higher frequency of malignant conversion to SCCs than WT papillomas (Figure 1). We have now compared cytokine levels in WT and EPI-/- SCCs (new Figure 8). There was no difference in the expression of IL-4 (a type-2 cytokine) or IL-17 (a type-17 cytokine) (Figure 8). However, EPI-/- SCCs expressed higher TSLP levels (Figure 8) and EPI-/- mice bearing papillomas larger than 2mm^2^ had higher levels of systemic TSLP than EPI-/- mice with smaller papillomas (Figure 8). Our data again highlight the difference between our model and that of Kopan, and corroborate reports that TSLP is a potent growth-promoting cytokine in breast and pancreatic cancers ([12]; [42], now cited).

*The reviewers felt that the link clinically between atopic dermatitis and skin cancer is weak, and here, some of this can be handled textually, by making the paper crisper in its focus and making fewer sweeping statements*.

We agree and we have revised the text accordingly.

Experimental issues/interpretations of the data:

*1) While EPI-/- mice have defects in skin tumor initiation, it is possible that these genes also play a role in tumor progression. Are Envoplakin, Periplakin or Involucrin expressed in WT skin papillomas or SCCs? If so, they may play additional roles once the skin tumor is already formed*.

We have included immunostaining for envoplakin, periplakin and involucrin in WT skin, papillomas and SCCs (new Figure 1), and show downregulation of all three proteins in tumours, which is most pronounced in SCCs. We agree that this could play a role in tumour progression as envoplakin and periplakin are desmosome-associated genes, and we now comment on this in the text.

*2) The authors nicely show that the mutagenic effect of DMBA is unaffected by the EPI mutations. The pro-inflammatory effect of TPA, however, is strongly exaggerated in EPI-/- mice. This is likely because barrier defect in these mice makes them more susceptible to the toxic effect of TPA. The authors note that there is increased expression of some inflammatory genes, genes involved in lymphoid stress immune-surveillance (e.g., RAE1) and barrier defense (e.g., TSLP). Blocking TSLP and NKG2D (receptor for RAE1) significantly reduce the inflammatory response. However, another group found that a reduction in DETC number resulted in increased skin tumorigenesis (Antisferova, Nature Communications, 2011). Although EPI-/- mice also have fewer DETCs, they develop fewer tumors. Could the overall increase in immune activation in EPI-/- mice render DETC function dispensable? Some comments on this difference would be helpful*.

We have now included images of γδ T cells (DETCs) in acetone treated EPI-/- and WT mice to show that in EPI-/- epidermis there are fewer γδ T cells (as we reported previously) and those that remain are in an activated state, as demonstrated by their rounded shape (new Figure 4). It has previously been reported that upon activation γδ T cells acquire a round shape before leaving the epidermis (Strid et al., Nature Immunology 2008; cited). K14-activin βA expressing mice (Antisferova et al., now cited) differ from EPI-/- mice, because downregulation of DETCs only occurs after DMBA/TPA treatment. The exaggerated TPA response in EPI-/- mice resulted in increased numbers of CD4+ T cells, neutrophils, mast cells and eosinophils, and many of these leukocyte populations have been shown to be able to exert tumor protective effects, which could potentially bypass the function of γδ T cells.

*3) Are there any differences in the presence*
*of alpha/beta T cells in EPI-/- epidermis?*

We have now included immunostaining for CD4+ T cells in TPA treated WT and EPI-/- skin, showing that CD4+ T cells are not present in WT epidermis, but do infiltrate EPI-/- epidermis (new Figure 4). In contrast, CD8+ T cells infiltrate the skin of untreated and TPA treated WT and EPI-/- mice to the same extent.

*4) The authors use antibodies against cytokines to suppress cytokine signaling during TPA treatment which many of these cytokines (in particular TSLP) have been implicated before. An extension of this mechanistic angle should implicate which cell types are responsible for the suppression of tumorigenesis. How these signals act in the immune cells etc*.

As described in the opening response, we have now included additional antibody targeting strategies (new Figure 6). One of the novel aspects of our work is that NKG2D has not been implicated before in linking atopic disease with skin tumorigenesis. Our studies show that blocking TSLP and NKG2D can normalize the exaggerated atopic response of keratinocytes and multiple leukocyte types in TPA treated EPI-/- skin, indicating that these targeting strategies affect a plethora of different cell types.

*5) A model might help this manuscript to better depict mechanistic findings of this manuscript. As presented the paper appears to be a set of fairly standard experiments with histology and RT-PCR*.

We have now included a model summarizing our findings (new Figure 9).

*6) The study reports two sets of findings: first, that EPI mice are protected from DMBA/TPA induced skin papillomas; second, that EPI mice have exaggerated inflammatory response to TPA. It is not clear whether the two are related and whether (and how) the latter explains the former. Normally, this would be tested using either mice deficient in the suspected components (TSLP, NKG2D), or by blocking these proteins with antibodies. The reviewers agreed that neither approach can be reasonably used here because the mice are already triple knockouts and because the treatment with antibodies for several months is not practical*.

Given that mice overexpressing TSLP in keratinocytes are protected from DMBA/TPA induced skin tumorigenesis (11), our data linking the exacerbated TPA response to upregulation of TSLP provides a strong link between our two sets of findings. We believe that this is clarified in the revised version of the manuscript and by inclusion of the model in Figure 9. Our results are of considerable interest because they provide experimental support for the emerging epidemiological data that allergic defenses can confer protection against environmental carcinogens in some contexts.

Textual issues/interpretations of the data:

*1) The authors may want to consult with an immunologist regarding their terminology and the choice of genes used as indicators of different types of immune responses. For example, IL-12p40 is not a Th17 marker and IL-6 is not a Th2 marker. These cytokines are made by myeloid cells and affect multiple responses; they are not produced by T cells*.

We have now changed the terminology to type-1, type-2, and type-17 responses (Figure 5).

*2) In addition, the article could benefit from some heavy editing. For example, the authors present as a fact that there is a causal relationship between barrier impairment, structural proteins, and atopic dermatitis. But the authors should use greater precision. In humans, mutations in FLG (a structural protein) are linked to ichthyosis vulgaris and associated with atopic dermatitis and atopic disorders, such as asthma and allergic rhinitis. But the authors state causality, when genetically correlation has been defined. Mouse models of FLG-/- display some features of AD, but not allergic airway inflammation (asthma model). Moreover these findings are more in question now that the animal model (flaky tail) has been shown to be a compound mutation of FLG and Tmem79*.

We have revised the text to address these points.

*There is true controversy that cancer is associated with atopy. Some references to support this model are Hwang CY (2012) Int J Cancer and Wedgeworth E (2011) Br JDerm but many dermatologists feel that the connection is more tenuous and may be driven by severe atopics being treated with heavy immunosuppressants, which have been associated with increased risk of malignancies. UV light therapy is also a potential treatment that would underlie an increase risk of skin cancer. These are confounding issues for patient data*.

We have revised the text and included these references.

*3) The extent to which DMBA/TPA treatment is a model for human skin cancer needs to be better addressed in the manuscript as most human SCC do not have H-ras mutations. These issues are mentioned, but typically toward the end of arguments, and need to be addressed before a more simplistic interpretation is depicted as the truth*.

We have now revised the Introduction to address this point and cited reviews by Lopez-Pajares et al. and Hirst and Balmain.

*4) Authors should read through and edit the entire manuscript. There are lots of circuitous arguments, typographical errors, and information provided in the wrong sections*.

We have subjected the text to extensive editing to address this criticism.

Methods/statistics/clarifications:

*1) How was the function of the antibody blockade of TSLP or NKG2D measured? What percent neutralization*
*was achieved?*

We demonstrate potent blockade of TSLP and IL-4 by ELISA (see Figures 6 and 7). We show reduced spleen mass as a readout for efficacy of Dexamethasone (new Figure 6) and show that anti-LY6G reduces both the number of neutrophils in circulation and the number of neutrophil containing skin pustules (new Figure 6). There is no straightforward method to confirm functional inhibition of NKG2D as this receptor has multiple downstream targets that are shared with other pathways. Nevertheless, we conclude that NKG2D function was at least partially blocked, because of the statistically significant differences in immune cell infiltration and epidermal thickness (Figures 7 and 8).

*2) How was RNA normalized for the acanthosis observed in EPI-/- skin? Basically, if RNA levels are higher, does this reflect an upregulation of the gene or simply that there are more suprabasal cells in the skin? Was RT-PCR deltaCt normalized*
*to B2u or some other ubiquitous gene?*

All normalizations were performed relative to GAPDH. We can exclude the possibility that QPCR data reveal differences in acanthosis rather than specific gene effects, as gene expression levels of Notch pathway genes, H60 and Rae-1 were similar in TPA treated (acanthotic) and vehicle treated EPI-/- epidermis (Figure 5).

*3) In*
Figure 3*, how is the percent of hyperkeratotic*
*and parakeratotic epidermis quantified?*

We quantified hyperkeratosis by measuring the length of epidermis that had marked thickening of the cornified layers relative to untreated WT epidermis. Parakeratotic regions were apparent from the persistence of nuclei in the cornified layers, which are normally anuclear. Parakeratotic regions were quantified by measuring the length of epidermis that was covered by parakeratotic plaques. We now explain this more clearly in the text.

*4) Are wt animals +/+;+/+;+/+ or compound heterozygotes. Nu/nu is not a genetic background but rather exists on multiple genetic backgrounds. This is confusing in the Methods section*.

We apologize for the confusion in the description of the genetic background of EPI-/- mice in the Methods section. WT animals used in this study were +/+:+/+;+/+. We checked for the presence of the Nu/Nu mutation on the BALBc background and all mice analyzed were negative. We have now made this section clearer.

*5) How representative are the*
*4 EPI-/- mice of genetic variation?*

To address this we performed SNP analysis on EPI-/- and WT mice to determine the relative contributions of the different genetic backgrounds. This showed that EPI-/- mice were primarily Sv129 (40.98%, ± 1.52) and C57Bl/6 (51.39%, ± 1.62). We therefore used F2 crosses between Sv129 and C57Bl/6 mice as the WT control (51.82%, ± 3.35 Sv129; 49.92%, ± 1.98 C57Bl/6) for all experiments, as now explained in the text. The 4 EPI-/- mice analyzed came from 4 separate breeding pairs that were used to generate the mice for the DMBA/TPA carcinogenesis experiments.